# Malaria treatment-seeking behaviour and its associated factors: A cross-sectional study in rural East Nusa Tenggara Province, Indonesia

**Robertus Dole Guntur**[1,2]*, **Jonathan Kingsley**[1,3], **Fakir M. Amirul Islam**[1]

**1** Department of Health Science and Biostatistics, Swinburne University of Technology, Hawthorn, Victoria, Australia, **2** Department of Mathematics, Faculty of Science and Engineering, Nusa Cendana University, Kupang, NTT, Indonesia, **3** Centre of Urban Transitions, Swinburne University of Technology, Hawthorn, Victoria, Australia

* rguntur@swin.edu.au

## Abstract

### Introduction

The World Health Organization recommends seeking medical treatment within 24 hours after transmission of malaria to reduce the risk of severe complications and its onwards spread. However, in some parts of Indonesia, including East Nusa Tenggara Province (ENTP), this adherence is not achieved for a range of reasons including delays in visiting health centres. This study aims to determine factors related to the poor understanding of appropriate malaria treatment-seeking behaviour (AMTSB) of rural adults in ENTP. AMTSB was defined as seeking treatment at professional health facilities within 24 hours of the onset of malaria symptoms.

### Methods

A cross-sectional study was conducted in the East Sumba, Belu, and East Manggarai district of ENTP between October and December 2019. A multi-stage cluster sampling procedure was applied to enrol 1503 participants aged between 18 and 89 years of age. Data were collected through face-to-face interviews. Multivariable logistic regression analyses were used to assess significant factors associated with the poor understanding of AMTSB.

### Results

Eighty-six percent of participants were found to be familiar with the term malaria. However, poor understanding level of AMTSB in rural adults of ENTP achieved 60.4% with a 95% confidence interval (CI): 56.9–63.8. Poor understanding of AMTSB was significantly higher for adults with no education (adjusted odds ratio (AOR) 3.42, 95% CI: 1.81, 6.48) compared to those with a diploma or above education level; having low SES (AOR: 1.87, 95% CI: 1.19, 2.96) compared to those having high SES; residing at least three kilometres (km) away from the nearest health facilities (AOR: 1.73, 95% CI: 1.2, 2.5) compared to those living within one km from the nearest health service; and working as farmer (AOR: 1.63, 95% CI: 1.01–2.63) compared to those working at government or non-government sector. Whilst, other

**Data Availability Statement:** All relevant data are within the manuscript and its Supporting Information files.

**Funding:** PhD scholarship for RDG was supported by the Australia Awards Scholarship (ST000TBK6). The Faculty of Health, Arts and Design (FHAD) of the Swinburne University Technology supported for the primary data collection. The funders had no role in study design, data collection and analysis, or interpretation of data or writing the manuscripts.

factors such as ethnicity and family size were not associated with the poor understanding of AMTSB.

## Conclusion

The proportion of rural adults having a poor understanding of AMTSB was high leading to ineffective implementation of artemisinin-based combination therapies as the method to treat malaria in ENTP. Improving awareness of AMTSB for rural adults having low level education, low SES, working as a farmer, and living at least three km from the nearest health facilities is critical to support the efficacy of malaria treatment in ENTP. This method will support the Indonesian government's objective to achieve malaria elimination by 2030.

## Introduction

The number of malaria cases in 2019 was estimated to be 229 million globally with a decrease of 4% in the last two decades [1]. This achievement was contributed to increasing coverage and access to various malaria control measures including the use of artemisinin-based combination therapies (ACTs) as the first-line method to treat malaria as recommended by the World Health Organization (WHO) [2–5]. The coverage of the use of ACTs increased significantly from 39% in the period from 2005 to 2011 to 81% in the period between 2015 and 2019 globally [1].

The effectiveness of this intervention depends greatly on various factors, including the behaviour of the community to seek timely treatment of the condition at health facilities [6–8]. An analysis of the malaria treatment-seeking behaviour database globally showed that treatment-seeking rates differ both within and among regions. For example, the rate of treatment-seeking behaviour of communities in the WHO Southeast Asia Region (SEARO) at government facilities, which provide comprehensive diagnostic and treatment schedule for malaria [2, 9] was only 27.6% (95% CI: 26.3–29.1), which is the lowest of all other WHO regional offices [10]. However, the rate of accessing any sources of treatment in SEARO communities was 78.8% (95%CI: 77.4–80.2), which was the highest proportion out of all countries reported in 2016 [10].

Time of treatment-seeking plays an important role in supporting the effectiveness of malaria management. The WHO recommends that treatment for malaria should occur within 24 hours of the onset of malaria symptoms to prevent the advancement of infection [2, 5, 11]. A recent systematic review on the impact of delaying in seeking malaria treatment indicates that the risk of severe malaria for patients seeking care more than 24 hours was higher than those seeking care less than 24 hours (odds ratio: 1.33, 95% CI: 1.07–1.64) [12]. Moreover, studies on various settings among Asia communities indicated that patients sought care for malaria at health facilities on average at least two days after the onset of symptoms [13–16]. One major reason of delaying in seeking malaria treatment of patients in rural communities of low to middle-income Asia-Pacific countries was the use of traditional medicine (TM) [17]. The prevalence of the application of TM as the first option before visiting health facilities in rural settings of the region varied from 2.5% to 40%. Other reasons for this included difficulty in accessing health facilities [16], living far from health facilities [13, 15, 16], and practicing self-treatment by buying drugs at local shops [14].

Indonesia remains the second largest contributor to the total number of malaria cases in South East Asia, with an estimate of 658,380 cases reported in 2019 [1]. Of this number, the

most common species of malaria parasites were *Plasmodium falciparum* and *Plasmodium vivax* accounting for 32% and 20% cases respectively [1]. Malaria treatment applying ACTs has been in use in Indonesia since 2004 [18–20]. This medicine is available for free of charge in all government health facilities throughout Indonesia [19, 20]. The coverage of malaria patients receiving ACTs to treat malaria has increased significantly over this period, especially in recent times, raised from 33.7% in 2013 [21] to 94.9% in 2020 [22]. This has been a part of the national effort to achieve malaria elimination by 2030 [19, 20, 23].

The high coverage of ACTs in Indonesia, however, is hampered by the behavior of people who treat their malaria without consultation from health practitioners and delay in seeking care at health facilities. The prevalence of self-medication for malaria treatment among the Indonesian population accounted for 0.6% of the populace, with variation from 0.2% in Bali Province to 5.1% in West Papua Province [21]. Furthermore, most of malaria patients in some parts of Indonesia visited health facilities after three days of the onset of symptoms [24–26] and the use of TM as the first option for malaria treatment leads to delay for several days in visiting health services [27, 28]. However, the associated factors related to delay in malaria treatment seeking behaviour of the patients were not investigated in aforementioned studies.

A review of the current database of health-seeking behaviour of the Indonesian population notes overall that the Indonesian population tends to defer the seeking of health care until the disease has worsened [29]. To date, the majority of health-seeking behaviour research for communicable diseases including malaria has been conducted in the Western part of Indonesia. For the eastern part of the country, including from the East Nusa Tenggara Province (ENTP), the literature is limited [29].

This is an omission given that ENTP, comprising of 10 main ethnicities [30] distributing across 624 islands in this archipelago province [31] is the province contributing the third largest malaria burden in Indonesia. The total number of positive cases reported in 2020 was 15,304 cases [22]. The proportion of malaria cases due to *Plasmodium vivax* in this province was higher compared to malaria cases because of *Plasmodium falciparum* [32, 33].

Limited investigations into malaria treatment-seeking behaviour had been conducted in the ENTP [21, 26, 34–36]. One study drawing on a small sample at the sub-district level indicated that most participants sought care at health facilities after four days of the onset of symptoms [26]. Two population studies covering both rural and urban populations indicated a high prevalence of residents seeking a cure by self-medication, accounting for 2.7% [21]. This also saw a high prevalence of the local community buying malarial medicines at a kiosk without consultation with health professionals [35]. Another study covering a Tetun community in Timor islands indicated the prevalence of community members applying various local medicinal plants to treat malaria [34]. One study representing the rural population in the province level indicated that more than half of the population delayed in seeking malaria treatment [36]. However, factors associated with this poor understanding of appropriate malaria treatment seeking behaviour (AMTSB) of rural communities have not been investigated yet. A better understanding of the factors contributed to the malaria health-seeking behaviour of the community has the potential to guide health policy makers in designing effective malaria treatment in the local context [37, 38]. To date there is no study at population level on the factors associated with the poor understanding of AMTSB covering various ethnicities in rural population in the ENTP. To fill this gap, we conduct this study to investigate factors associated with the poor understanding of malaria treatment-seeking behaviour of rural community in the province to support national commitment to achieve zero local transmission by 2030.

## Methods

### Study area

A cross-sectional study was conducted in ENTP from October to December 2019. The province comprising of five main islands including Sumba, Flores, Timor, Alor, and Lembata has a total population of 5,456,203 with the ratio of males to females being 49.5% versus 50.5% [31]. Data collection was conducted in Sumba Island which makes 70% of the total malaria cases in this province (Timor Island representing 10% and Flores island 6% of the total malaria cases in this province) [39].

Health care providers in this province are mainly supported by the government sector with state hospitals at district and province level and public health centre (PHC) at sub-district level, which is locally known as "Puskesmas". When this research was carried out, the total number of PHC is 381 distributing across 309 sub-districts where the infrastructure is limited including no electricity in some PHCs [40]. The distribution of general practitioner (GP) in each PHC is uneven ranging from zero to ten with 47% of the total number of PCH was supported with one GP and 33% of them have no availability [40]. Following the guideline of the national malaria control program of the Indonesian government, the diagnosis of malaria cases in PHC was confirmed with laboratory tests or rapid diagnostic tests and the treatment for malaria with applied ACT as the first-line treatment [39, 41]. Malaria patients caused by *Plasmodium falciparum* were treated with ACTs for three days and primaquine for one day, whilst malaria patients caused by *Plasmodium vivax* were treated with ACTs for 3 days and primaquine for fourteen days [41, 42].

### Sample size and recruitment

The total sample recruited for the study was 1503 adults. This number was obtained after considering the prevalence of malaria in ENTP, the intra-class coefficient correlation for malaria prevention study in Indonesia, design effect, and participation rate of participants. The complete sample size calculation was presented previously [43].

A multi-stage random cluster sampling was conducted to obtain data from 49 clusters in the rural village of ENTP Indonesia. The description of sampling procedure was described previously [43]. Overall, three districts were selected based on their malaria-endemic settings (MES), which were East Manggarai, Belu, and East Sumba district, representing as low, moderate, and high MES [44]. In each district, three sub-districts were selected randomly. The number of a cluster were selected in each selected district according to the size of the population. Since the size of each cluster was different, we selected 25 to 40 participants in each cluster proportion to the cluster size. The selection of this participants in each cluster applied a systematic random sampling method. One household head was selected as study participant in the visited households. Any household head whether male or female who wanted to participate had to provide a voluntary consent form and be over the age of 18 years to be included in this study. In cases when there were two or more household heads in single dwelling, the household head occupying firstly on that home was selected as participants. Since we want to explore malaria treatment seeking behavior of rural adults in ENTP, we exclude any household heads who were less than 18 years of age. This criterion was based on the guidance of the ethic of this research.

### Data collection

Data collection was conducted by local nurses who participated in a one-day intensive training in the capital city of selected districts before conducting the survey. The local language was

used to interview participants face-to-face based on the questions that had been provided in a validated questionnaire. Data on socio-demographic characteristics and environmental variables were collected including access to a water tap in the household, and the ownership of items of durable assets including radio, television, hand phone, motorbike, bike, electricity, fridge, generator, tractor, car, and modern house. These data were used to construct the social-economic status of participants. The comprehensive questionnaire has been published by the authors previously [43]. Three main questions as indicated in Table 1 were asked to participants to capture their understanding on malaria treatment-seeking behaviour.

## Outcome variables

There were three main outcomes of the study including perceptions on: (i) seeking malaria treatment after 24 hours, (ii) pursuing treatment at non-health facilities and, (iii) the poor understanding of appropriate malaria treatment-seeking behaviour (AMTSB). All these outcomes were dichotomous variables having categories yes which was marked as one or no that was marked as zero. The AMTSB was defined as seeking treatment at professional health facilities within 24 hours of the onset of the malaria symptom [2, 45]. Based on the response of participants to the three main questions, they were categorized into three groups. The first group was the participants who sought treatment after 24 hours of detection of the malaria symptoms. The second group was the participants who sought treatment at non-health facilities. The third group comprised participants who sought treatment after 24 hours or at non-health facilities. All participants categorized as the third group were defined as having a poor understanding of AMTSB [46].

## Independent variables

There are a range of socio-demographic and environmental covariates that have shown the association with care-seeking malaria treatment. These include gender [46]; age group [46, 47]; education level [46, 48, 49]; social-economic status (SES) [50–52]; occupation [53, 54]; income [15]; family size [47]; types of health facilities [45]; distance to the health facilities [13, 15, 55]; and ethnicities [56]. In this study, gender was classified as males or females. The age group was divided into five categories: < 30 years, 30–39 years, 40–49 years, 50–59 years, and ≥ 60 years old. The education level of participants was classified as no education, primary education, junior education, senior education, and diploma or above education level. The main

**Table 1. Questions used to explore understanding of treatment-seeking behaviour of participants.**

| Questions | The possibility response of participants |
|---|---|
| Would you find treatment if you or your family members were suffering from malaria? | 1. Yes |
| | 2. No |
| How fast will you try to find treatment if you or your family is affected by malaria | 1. Within 24 hours |
| | 2. Two days |
| | 3. Three days |
| | 4. Four days |
| | 5. I did not seek treatment |
| Where would you try to find treatment for your malaria | 1. Public health centres |
| | 2. Private health centres |
| | 3. Traditional healers |
| | 4. Self-treatment |
| | 5. Buying medicine at the kiosk |

occupation of participants was categorized as a farmer, housewife, entrepreneur, government and non-government workers, and other occupations. The type of health facilities was classified as village maternity posts, village health posts, public health centres (PHC), and subsidiary PHC. The distance to the closest health facilities was defined as < 1 kilometres (km), 1–2 km, 2–3 km, ≥3 km. The ethnicities comprised Manggarai, Sumba, Atoni, and other ethnicities. The SES of participants was classified as high, moderate, and low as indicated in the previous publication of the authors [36].

### Data analysis

Participant's characteristics including gender, age group, education level, main occupation, ethnicity, the nearest health service, and the distance to the nearest health service were analysed by descriptive statistics. The proportion of participants based on the first, second, and third outcome variables were tabulated based on the socio-demographic and environmental variables of participants. The association between outcomes and various independent variables was investigated by the chi-square test. Logistic regression methods were conducted to investigate the strengths of association of each independent variable with the outcome variables. The crude odds ratio (OR) and the adjusted OR with 95% confidence intervals were demonstrated as the result of logistic regression analysis. Any p values $\leq$ 0.05 was considered to be significant.

### Ethical consideration

This study was carried out in line with the principle of the Declaration of Helsinki to ensure that the rights, integrity, and privacy of participants were firmly treated. The ethical approval was obtained from the Swinburne University of Technology Human Ethics Committee with reference: 20191428–1490 and the Health Research Ethics Committee of Indonesia Health Ministry with reference: LB.02.01/2/KE.418/2019. All participants provided written consent prior to interviews being conducted.

## Results

The total number of participants interviewed in this study was 1495 adults. Of the total participants, 86% had heard malaria term, of which 88.7% were male, 82.6% finished their primary school, 88.1% were farmers, 82% were from low SES, and 84.1% were from a household with a family member of four or less. Thirteen point nine percent (208 out of 1495) of participants have never heard of malaria. Of this number, 16.4% were female, 17.4% were completed their primary school, 21.1% were from Atoni ethnicity, and 16.3% lived less than one kilometres from the nearest health facilities as shown in Fig 1.

### Perception on finding malaria treatment

Perception of time and place to seek malaria treatment of rural adults of ENTP is presented in Table 2. Of 1287 participants hearing malaria term, almost all of them 98.8%, 95% confidence interval (CI): 98.2–99.4 reported to find malaria treatment if they or their family members suffering from malaria, and 82.4%, 95% CI: 80.2–84.7 sought treatment at public health facilities, however, only 46.8%, 95% CI: 42.8–50.8 of participants sought treatment within 24 hours of the onset of the malaria symptoms. Overall, 53.2%, 95% CI: 49.5–57.0, of participants had a perception to seek malaria treatment after 24 hours, and 15.7%, 95% CI: 10.7–20.7, believed to find treatment in non-health facilities. Overall, the percentage of participants having a poor understanding of AMTSB accounted for 60.4% with 95% CI: 56.9–63.8.

| Characteristic | Hearing Malaria Term | | | |
| --- | --- | --- | --- | --- |
| | Yes | | No | |
| | n | % | n | % |
| Overall | 1287 | 86.1 | 208 | 13.9 |
| **Gender** | | | | |
| Female | 642 | 83.6 | 126 | 16.4 |
| Male | 645 | 88.7 | 82 | 11.3 |
| **Age group** | | | | |
| < 30 | 185 | 90.2 | 20 | 9.8 |
| 30 − 39 | 374 | 89.5 | 44 | 10.5 |
| 40 − 49 | 333 | 89.8 | 38 | 10.2 |
| 50 − 59 | 243 | 82.4 | 52 | 17.6 |
| ≥ 60 | 152 | 73.8 | 54 | 26.2 |
| **District** | | | | |
| East Sumba | 480 | 97.0 | 15 | 3.00 |
| Belu | 398 | 79.6 | 102 | 20.4 |
| East Manggarai | 409 | 81.8 | 91 | 18.2 |
| **Level of education** | | | | |
| No Education | 232 | 83.2 | 47 | 16.8 |
| Primary School | 560 | 82.6 | 118 | 17.4 |
| Junior High School | 206 | 90.0 | 23 | 10.0 |
| Senior High School | 194 | 92.4 | 16 | 7.60 |
| Diploma or above | 95 | 96.0 | 4 | 4.00 |
| **Ethnicities** | | | | |
| Sumba | 439 | 97.1 | 13 | 2.90 |
| Others | 63 | 94.0 | 4 | 6.00 |
| Atoni | 381 | 78.9 | 102 | 21.1 |
| Manggarai | 404 | 81.9 | 89 | 18.1 |
| **Main Occupation** | | | | |
| Farmer | 732 | 88.1 | 99 | 11.9 |
| Housewife | 311 | 77.2 | 92 | 22.8 |
| Entrepreneur | 43 | 89.6 | 5 | 10.4 |
| Other | 60 | 96.8 | 2 | 3.20 |
| Govt. or non-govt. employment | 141 | 93.4 | 10 | 6.60 |
| **Socio-Economic Status** | | | | |
| Low | 368 | 82.0 | 81 | 18.0 |
| Average | 746 | 86.7 | 114 | 13.3 |
| High | 173 | 93.0 | 13 | 7.00 |
| **Family size** | | | | |
| ≤ 4 | 675 | 84.1 | 128 | 15.9 |
| > 4 | 612 | 88.4 | 80 | 11.6 |
| **The nearest Health Service** | | | | |
| Village maternity posts | 337 | 87.3 | 49 | 12.7 |
| Village health Post | 226 | 74.8 | 76 | 25.2 |
| Subsidiary Public Health centres | 313 | 92.6 | 25 | 7.40 |
| Public Health centres | 411 | 87.6 | 58 | 12.4 |
| **Distance to the Nearest Health Facilities** | | | | |
| < 1 Km | 484 | 83.7 | 94 | 16.3 |
| 1 − 2 Km | 360 | 90.0 | 40 | 10.0 |
| 2 − 3 Km | 174 | 85.3 | 30 | 14.7 |
| ≥ 3 Km | 269 | 85.9 | 44 | 14.1 |
| **†HH Income in relation to PMW** | | | | |
| < PMW | 1151 | 85.8 | 191 | 14.2 |
| >= PMW | 136 | 88.9 | 17 | 13.9 |
| **‡HH Income in relation to IPL** | | | | |
| < IPL | 1199 | 85.9 | 196 | 14.1 |
| ≥ IPL | 88 | 88.0 | 12 | 12.0 |

†The Provincial Minimum Wages (PMW) in 2019 is IDR 1,795,000 monthly; ‡The Indonesia Poverty Line (IPL) in 2019 is defined all households having the average income less than IDR 1,990,170 monthly.

**Fig 1. Awareness of malaria and its association with the sociodemographic and environmental characteristics of respondents in the East Nusa Tenggara Province, Indonesia (n = 1495).**

## Seeking malaria treatment beyond 24 hours and its factors associated

The distribution of participants seeking malaria treatment after 24 hours and their factors associated were presented in Fig 2. Overall, 53.2% of participants had the perception to seek

**Table 2. Perception on finding treatment if respondents or their family members have any symptoms of malaria of people who were aware of malaria (n = 1287).**

|  | n | % | 95% CI |
|---|---|---|---|
| **Would you find treatment if you or your family is affected by malaria?** | | | |
| Yes | 1271 | 98.8 | (98.2, 99.4) |
| No | 16 | 1.20 | (0.00, 6.67) |
| **How fast would you try to find treatment if you or your family has malaria?** | | | |
| One day (Within 24 hours) | 602 | 46.8 | (42.8, 50.8) |
| 2 days | 306 | 23.8 | (19.0, 28.6) |
| 3 days | 286 | 22.2 | (17.4, 27.0) |
| 4 days or more | 70 | 5.40 | (0.13, 10.8) |
| I did not go for treatment | 23 | 1.80 | (0.00, 7.23) |
| **Where would you try to find the treatment** | | | |
| Public health facilities | 1061 | 82.4 | (80.2, 84.7) |
| Private health facilities | 24 | 1.90 | (0.00, 7.28) |
| Traditional healer | 102 | 7.90 | (2.68, 13.2) |
| Self-treatment | 48 | 3.70 | (0.00, 9.04) |
| Buying medicine at kiosk | 44 | 3.40 | (0.00, 8.79) |
| Self-treatment with consuming papaya leaves | 8 | 0.60 | (0.00, 6.07) |
| Perception on finding treatment after 24 hours | 685 | 53.2 | (49.5, 57.0) |
| Perception on finding treatment in non-health facilities | 202 | 15.7 | (10.7, 20.7) |
| Poor understanding on the appropriate malaria treatment seeking behaviour (participants having perception on finding treatment after 24 hours or in non-health facilities) | 777 | 60.4 | (56.9, 63.8) |

malaria treatment after 24 hours. Compared to 44.5% of participants living closer to public health centres, 63.7% of participants resided closer to village health posts had the perception to seek malaria treatment after 24 hours. The highest percentage of seeking malaria treatment beyond 24 hours was in participants with no education level (69.8%), whilst the lowest was in participants with a diploma or above education level (27.4%).

After adjustment for gender and age group, the following variables were significantly associated with the perception of seeking malaria treatment after 24 hours: no education level compared to those with a diploma or above education level (Adjusted Odds ratio (AOR) 3.49, 95% confidence interval (CI) 1.86–6.55,); poor SES compared to those were from high SES (AOR: 2.24, 95% CI: 1.44–3.47,); village health post compared to those were living close to the public health centre (AOR: 2.43, 95% CI: 1.66–3.54); living more than three kilometres compared to those were living less than one kilometres from health facilities (AOR: 1.65, 95% CI: 1.16–2.33); household income less than provincial minimum wages (PMW) compared to those were from the household with income more than PMW (AOR: 2.13, 95%CI: 1.36–3.32).

| Characteristics | No at risk | Perception on seeking treatment beyond than 24 hours | | | |
| --- | --- | --- | --- | --- | --- |
| | | n (%) | COR (95% CI)* | AOR (95% CI)** | P value |
| Overall | 1287 | 685 (53.2) | | | |
| **Gender** | | | | | |
| Female | 642 | 349 (54.4) | 1.10 (0.88, 1.36) | | |
| Male | 645 | 336 (52.1) | 1.00 (Reference) | | |
| **Age group** | | | | | |
| < 30 | 185 | 106 (57.3) | 0.70 (0.49, 1.00) | | |
| 30 – 39 | 374 | 181 (48.4) | 0.93 (0.65, 1.34) | | |
| 40 – 49 | 333 | 185 (55.6) | 0.75 (0.51, 1.10) | | |
| 50 – 59 | 243 | 122 (50.2) | 1.11 (0.72, 1.72) | | |
| ≥ 60 | 152 | 91 (59.9) | 1.00 | | |
| **District** | | | | | |
| East Sumba | 480 | 313 (65.2) | 3.17 (2.41, 4.17) | 1.37 (0.99, 1.89) | 0.058 |
| Belu | 398 | 220 (55.3) | 2.09 (1.58, 2.77) | 0.54 (0.38, 0.76) | 0.000 |
| East Manggarai | 409 | 152 (37.2) | 1.00 | 1.00 | |
| **Level of Education** | | | | | |
| No Education | 232 | 162 (69.8) | 6.14 (3.61, 10.5) | 3.49 (1.86, 6.55) | 0.000 |
| Primary School | 560 | 312 (55.7) | 3.34 (2.06, 5.40) | 2.50 (1.43, 4.37) | 0.001 |
| Junior High School | 206 | 86 (41.7) | 1.90 (1.12, 3.23) | 1.45 (0.80, 2.62) | 0.225 |
| Senior High School | 194 | 99 (51.0) | 2.77 (1.63, 4.71) | 2.38 (1.32, 4.26) | 0.004 |
| Diploma or above | 95 | 26 (27.4) | 1.00 | 1.00 | |
| **Ethnicities** | | | | | |
| Sumba | 439 | 296 (67.4) | 3.47 (2.61, 4.61) | 2.30 (0.51, 11.9) | 0.305 |
| Others | 63 | 29 (46.0) | 1.43 (0.84, 2.44) | 0.90 (0.20, 4.20) | 0.929 |
| Atoni | 381 | 209 (54.9) | 2.04 (1.53, 2.71) | 0.70 (0.10, 3.40) | 0.626 |
| Manggarai | 404 | 151 (37.4) | 1.00 | 1.00 | |
| **Main Occupation** | | | | | |
| Housewife | 311 | 161 (51.8) | 2.08 (1.38, 3.14) | 0.89 (0.49, 1.60) | 0.689 |
| Farmer | 732 | 413 (56.4) | 2.51 (1.72, 3.66) | 1.32 (0.76, 2.28) | 0.323 |
| Entrepreneur | 43 | 24 (55.8) | 2.45 (1.22, 4.91) | 0.98 (0.43, 2.23) | 0.957 |
| Other | 60 | 39 (65.0) | 3.60 (1.91, 6.79) | 1.82 (0.86, 3.84) | 0.118 |
| Govt. or non-govt. employment | 141 | 48 (34.0) | 1.00 | 1.00§ | |
| **Socio-Economic Status** | | | | | |
| Poor | 368 | 216 (58.7) | 2.25 (1.55, 3.25) | 2.24 (1.44, 3.47) | 0.000 |
| Moderate | 746 | 402 (53.9) | 1.85 (1.32, 2.59) | 1.61 (1.10, 2.36) | 0.013 |
| Rich | 173 | 67 (38.7) | 1.00 | 1.00 | |
| **Family size** | | | | | |
| ≤ 4 | 675 | 350 (51.9) | 1.12 (0.90, 1.40) | | |
| > 4 | 612 | 335 (54.7) | 1.00 | | |
| **The nearest Health Service** | | | | | |
| Village maternity posts | 337 | 211 (62.6) | 2.09 (1.55, 2.80) | 1.58 (1.13, 2.21) | 0.007 |
| Village health Post | 226 | 144 (63.7) | 2.19 (1.57, 3.05) | 2.43 (1.66, 3.54) | 0.000 |
| Subsidiary Public Health centres | 313 | 147 (47.0) | 1.1 (0.82, 1.48) | 0.81 (0.58, 1.13) | 0.218 |
| Public Health centres | 411 | 183 (44.5) | 1.00 | 1.00 | |
| **Distance to the Nearest Health Facilities** | | | | | |
| < 1 Km | 484 | 275 (56.8) | 1.00 | 1.00 | |
| 1 – 2 Km | 360 | 161 (44.7) | 0.61 (0.47, 0.81) | 0.82 (0.61, 1.12) | 0.212 |
| 2 – 3 Km | 174 | 69 (39.7) | 0.50 (0.35, 0.71) | 0.63 (0.43, 0.92) | 0.018 |
| ≥ 3 Km | 269 | 180 (66.9) | 1.54 (1.13, 2.10) | 1.65 (1.16, 2.33) | 0.005 |
| †HH Income in relation to PMW | | | | | |
| < PMW | 1151 | 645 (56.0) | 3.06 (2.08, 4.50) | 2.13 (1.36, 3.32) | 0.001 |
| >= PMW | 136 | 40 (29.4) | 1.00 | 1.00 | |
| ‡HH Income in relation to IPL | | | | | |
| < IPL | 1199 | 660 (55.0) | 3.09 (1.92, 4.97) | 0.95 (0.41, 2.20) | 0.911 |
| ≥ IPL | 88 | 25 (28.4) | 1.00 | 1.00§ | |

†The Provincial Minimum Wages (PMW) in 2019 is IDR 1,795,000 monthly; ‡The Indonesia Poverty Line (IPL) in 2019 is defined all households having the average income less than IDR 1,990,170 monthly; *Crude Odd Ratio (COR); **Adjusted Odd Ratio (AOR); §Variables are not significant at confidence interval (CI) 95%.

**Fig 2. The level of perception on seeking malaria treatments beyond 24 hours by different sociodemographic and environmental characteristics in the East Nusa Tenggara Province, Indonesia (n = 685).**

## Seeking malaria treatment at non-health facilities and its factors associated

The distribution of participants seeking malaria treatment at non-health facilities and their factors associated was presented in Fig 3. Of the total participants hearing malaria term, 15.7% had the perception to seek malaria treatment at non-health facilities. Compared to 11.7% of participants living closer to public health centres, 20.8% among those residing closer to village maternity post had the perception to seek malaria treatment at non-health facilities. The percentage of seeking malaria treatment at non-health facilities for those having no education was

| Characteristics | No at risk | Perception on seeking treatment at non health facilities | | | |
|---|---|---|---|---|---|
| | | n (%) | COR (95% CI)* | AOR (95% CI)** | P value |
| Overall | 1287 | 202 (15.7) | | | |
| **Gender** | | | | | |
| Female | 642 | 85 (13.2) | 1.00 (Reference) | 1.00 | |
| Male | 645 | 117 (18.1) | 1.45 (1.07, 1.97) | 1.44 (1.04, 1.98) | 0.026 |
| **Age group** | | | | | |
| < 30 | 185 | 24 (13.0) | 1.00 | | |
| 30 – 39 | 374 | 38 (10.2) | 0.76 (0.44, 1.31) | | |
| 40 – 49 | 333 | 57 (17.1) | 1.39 (0.83, 2.32) | | |
| 50 – 59 | 243 | 54 (22.2) | 1.92 (1.13, 3.24) | | |
| ≥ 60 | 152 | 29 (19.1) | 1.58 (0.88, 2.85) | | |
| **District** | | | | | |
| East Sumba | 480 | 68 (14.2) | 1.00 | 1.00 | |
| Belu | 398 | 57 (14.3) | 1.01 (0.69, 1.48) | 2.47 (1.59, 3.84) | 0.000 |
| East Manggarai | 409 | 77 (18.8) | 1.41 (0.98, 2.01) | 1.01 (0.66, 1.53) | 0.972 |
| **Level of Education** | | | | | |
| No Education | 232 | 46 (19.8) | 4.45 (1.71, 11.6) | 3.72 (1.29, 10.7) | 0.015 |
| Primary School | 560 | 111 (19.8) | 4.45 (1.77, 11.2) | 2.83 (1.04, 7.65) | 0.041 |
| Junior High School | 206 | 23 (11.20) | 2.26 (0.83, 6.15) | 1.53 (0.53, 4.40) | 0.433 |
| Senior High School | 194 | 17 (8.80) | 1.73 (0.62, 4.84) | 1.35 (0.46, 3.91) | 0.583 |
| Diploma or above | 95 | 5 (5.30) | 1.00 | 1.00 | |
| **Ethnicities** | | | | | |
| Sumba | 439 | 62 (14.1) | 1.02 (0.69, 1.51) | | |
| Others | 63 | 11 (17.5) | 1.31 (0.64, 2.67) | | |
| Manggarai | 404 | 76 (18.8) | 1.43 (0.98, 2.10) | | |
| Atoni | 381 | 53 (13.9) | 1.00§ | | |
| **Main Occupation** | | | | | |
| Housewife | 311 | 34 (10.9) | 3.34 (1.28, 8.73) | 2.06 (0.71, 5.99) | 0.184 |
| Farmer | 732 | 147 (20.1) | 6.83 (2.75, 17.0) | 3.63 (1.34, 9.85) | 0.011 |
| Entrepreneur | 43 | 5 (11.6) | 3.58 (0.98, 13.0) | 2.87 (0.73, 11.2) | 0.130 |
| Other | 60 | 11 (18.3) | 6.11 (2.02, 18.5) | 4.12 (1.27, 13.4) | 0.018 |
| Govt. or non-govt. employment | 141 | 5 (3.50) | 1.00 | 1.00 | |
| **Socio-Economic Status** | | | | | |
| Poor | 368 | 78 (21.2) | 2.64 (1.49, 4.68) | 1.34 (0.70, 2.55) | 0.371 |
| Moderate | 746 | 108 (14.5) | 1.66 (0.96, 2.89) | 1.08 (0.59, 1.96) | 0.804 |
| Rich | 173 | 16 (9.20) | 1.00 | 1.00§ | |
| **Family size** | | | | | |
| ≤ 4 | 675 | 107 (15.9) | 1.00§ | | |
| > 4 | 612 | 95 (15.5) | 1.03 (0.76, 1.39) | | |
| **The nearest Health Service** | | | | | |
| Village maternity posts | 337 | 70 (20.8) | 1.98 (1.33, 2.96) | 2.49 (1.58, 3.91) | 0.000 |
| Village health Post | 226 | 34 (15.0) | 1.34 (0.83, 2.15) | 1.12 (0.68, 1.85) | 0.652 |
| Subsidiary Public Health centres | 313 | 50 (16.0) | 1.44 (0.94, 2.20) | 1.25 (0.80, 1.95) | 0.337 |
| Public Health centres | 411 | 48 (11.7) | 1.00 | 1.00 | |
| **Distance to the Nearest Health Facilities** | | | | | |
| < 1 Km | 484 | 62 (12.8) | 1.00 | 1.00 | |
| 1 - 2 Km | 360 | 67 (18.6) | 1.56 (1.07, 2.27) | 0.74 (0.46, 1.17) | 0.193 |
| 2 - 3 Km | 174 | 31 (17.8) | 1.48 (0.92, 2.36) | 1.09 (0.69, 1.73) | 0.709 |
| >=3 Km | 269 | 42 (15.6) | 1.26 (0.82, 1.92) | 1.10 (0.63, 1.89) | 0.743 |
| **†HH Income in relation to PMW** | | | | | |
| < PMW | 1151 | 196 (17) | 4.45 (1.93, 10.2) | 3.24 (1.33, 7.94) | 0.010 |
| >= PMW | 136 | 6 (4.40) | 1.00 | 1.00 | |
| **‡HH Income in relation to IPL** | | | | | |
| < IPL | 1199 | 199 (16.6) | 5.64 (1.77, 18.0) | 1.73 (0.32, 9.41) | 0.483 |
| >= IPL | 88 | 3 (3.40) | 1.00 | 1.00§ | |

†The Provincial Minimum Wages (PMW) in 2019 is IDR 1,795,000 monthly; ‡The Indonesia Poverty Line (IPL) in 2019 is defined all households having the average income less than IDR 1,990,170 monthly; *Crude Odd Ratio (COR); **Adjusted Odd Ratio (AOR); §Variables are not significant at confidence interval (CI) 95%.

**Fig 3. The level of perception on seeking malaria treatment in non-health facilities by different sociodemographic and environmental characteristics in the East Nusa Tenggara Province, Indonesia (n = 202).**

the highest (19.8%) compared to only 5.30% for those having a diploma or above education level.

After adjustment for gender and age group, the following variables were significantly associated with the perception of seeking malaria treatment at non-health facilities: male compared to female (Adjusted Odds ratio (AOR) 1.44, 95% confidence interval (CI) 1.04–1.98); Belu

district compared to East Sumba district (AOR: 2.47, 95% CI: 1.59–3.84); no education level compared to those with a diploma or above education level (AOR: 3.72, 95% CI: 1.29–10.72); farmer compared to those with government or non-government workers (AOR: 3.63, 95% CI: 1.34–9.85); village maternity post compared to those were living close to the public health centre (AOR: 2.49, 95% CI: 1.58–3.91); household income less than provincial minimum wages (PMW) compared to those were from the household with income more than PMW (AOR: 3.24, 95%CI: 1.33–7.94).

## The poor understanding of appropriate malaria treatment-seeking behaviour (AMTSB) and its factors associated

The distribution of the poor understanding of AMTSB and its factors associated was presented in Fig 4. Overall, 60.4% of participants hearing malaria term had a poor understanding of AMTSB. The proportion of poor understanding was the highest (73.7%) for those having no education compared to those with a diploma or above education level (31.6%). Seventy one percent of participants with poor SES compared to 42.8% of participants with rich SES were found to have a poor understanding of AMTSB. More than half of participants in all ethnicities in the ENTP had a poor understanding of AMTSB with the highest in Sumba ethnicity (69.9%) and the lowest in other ethnicities (50.8%).

After adjustment for gender and age group, the following variables were significantly associated with the poor understanding of AMTSB for rural adults in ENTP: no education level compared to those with a diploma or above education level (Adjusted Odds ratio (AOR) 3.42, 95% confidence interval (CI) 1.81–6.48); household income less than provincial minimum wages (PMW) compared to those were from the household with income more than PMW (AOR: 2.08, 95%CI: 1.33–3.26); poor SES compared to rich SES (AOR: 1.87, 95% CI: 1.19–2.96); village health post compared to those were living close to the public health centre (AOR: 2.66, 95% CI: 1.80–3.94); participants living at least three kilometres away from the nearest health facilities compared to those living within one kilometre from the nearest health facilities (AOR: 1.73, 95% CI: 1.2–2.5); occupation as farmer compared to those working at the government or non-government sector (AOR: 1.63, 95%CI: 1.01–2.63).

## Discussion

This study addresses a research gap around understanding of AMTSB and its associated factors in rural communities in ENTP Indonesia. The main finding of this study shows that more than half of rural communities in ENTP have a perception to delay in malaria treatment-seeking behaviour and more than half of them have a poor understanding of AMTSB which leads to a significant obstacle to address malaria. The main factors associated with the poor understanding of the AMTSB were the low level of education, low social-economic status, farming occupation, and distance to the nearest health facilities.

This study shows that more than half of participants delay in seeking treatment for their malaria. This finding is consistent with studies in Myanmar [46] and Nigeria [47]. However, our finding contrasted with studies in Cabo Verde [57] and South Africa [58], revealing that the high awareness of the community to seek treatment within 24 hours. This discrepancy might be due to different level of malaria knowledge of communities in those settings. Rural communities in ENTP Indonesia [36], Myanmar [46] and Nigeria [47] has low malaria knowledge, whilst rural adults in Cabo Verde and South Africa have high malaria knowledge leads to timely seek malaria treatment [57, 58]. The low awareness to seek treatment within 24 hours leads to ineffective implementation of ACTs as the first line of malaria treatment, even though the coverage of ACTs in ENTP has increased from 55% in 2013 [21] to 93.9% in 2020 [22].

| Characteristics | No at risk | Poor understanding of the appropriate malaria treatment seeking behaviour | | | |
|---|---|---|---|---|---|
| | | n (%) | COR (95% CI)* | AOR (95% CI)** | P value |
| **Overall** | 1287 | 777 (60.4) | | | |
| **Gender** | | | | | |
| Female | 642 | 389 (60.6) | 1.02 (0.81, 1.27) | | |
| Male | 645 | 388 (60.2) | 1.00 (Reference) | | |
| **Age group** | | | | | |
| < 30 | 185 | 117 (63.2) | 1.00 | | |
| 30 – 39 | 374 | 203 (54.3) | 0.69 (0.48, 0.99) | | |
| 40 – 49 | 333 | 209 (62.8) | 0.98 (0.68, 1.42) | | |
| 50 – 59 | 243 | 148 (60.9) | 0.91 (0.61, 1.34) | | |
| > 60 | 152 | 100 (65.8) | 1.12 (0.71, 1.75) | | |
| **District** | | | | | |
| East Sumba | 480 | 326 (67.9) | 1.75 (1.33, 2.30) | 1.3 (0.92, 1.84) | 0.135 |
| Belu | 398 | 227 (57.0) | 1.10 (0.83, 1.45) | 0.98 (0.69, 1.39) | 0.916 |
| East Manggarai | 409 | 224 (54.8) | 1.00 | 1.00§ | |
| **Level of Education** | | | | | |
| No Education | 232 | 171 (73.7) | 6.07 (3.60, 10.2) | 3.42 (1.81, 6.48) | 0.000 |
| Primary School | 560 | 370 (66.1) | 4.22 (2.65, 6.73) | 2.58 (1.47, 4.53) | 0.001 |
| Junior High School | 206 | 99 (48.1) | 2.00 (1.20, 3.34) | 1.35 (0.75, 2.44) | 0.314 |
| Senior High School | 194 | 107 (55.2) | 2.66 (1.59, 4.47) | 2.20 (1.24, 3.91) | 0.007 |
| Diploma or above | 95 | 30 (31.6) | 1.00 | 1.00 | |
| **Ethnicities** | | | | | |
| Sumba | 439 | 307 (69.9) | 1.91 (1.44, 2.53) | 1.98 (0.40, 9.79) | 0.400 |
| Others | 63 | 32 (50.8) | 0.85 (0.50, 1.44) | 0.89 (0.21, 3.82) | 0.872 |
| Atoni | 381 | 216 (56.7) | 1.07 (0.81, 1.42) | 0.64 (0.13, 3.13) | 0.580 |
| Manggarai | 404 | 222 (55.0) | 1.00 | 1.00§ | |
| **Main Occupation** | | | | | |
| Housewife | 311 | 166 (53.4) | 1.96 (1.30, 2.95) | 0.84 (0.50, 1.43) | 0.528 |
| Farmer | 732 | 496 (67.8) | 3.60 (2.47, 5.24) | 1.63 (1.01, 2.63) | 0.046 |
| Entrepreneur | 43 | 24 (55.8) | 2.16 (1.08, 4.32) | 1.14 (0.52, 2.50) | 0.739 |
| Other | 60 | 39 (65.0) | 3.18 (1.69, 5.98) | 2.03 (1.00, 4.09) | 0.049 |
| Govt. or non-govt. employment | 141 | 52 (36.9) | 1.00 | 1.00 | |
| **Socio-Economic Status** | | | | | |
| Poor | 368 | 260 (70.7) | 3.22 (2.21, 4.69) | 1.87 (1.19, 2.96) | 0.007 |
| Moderate | 746 | 443 (59.4) | 1.96 (1.40, 2.73) | 1.35 (0.92, 1.99) | 0.128 |
| Rich | 173 | 74 (42.8) | 1.00 | 1.00 | |
| **Family size** | | | | | |
| ≤ 4 | 675 | 394 (58.4) | 1.00§ | | |
| > 4 | 612 | 383 (62.6) | 1.19 (0.95, 1.49) | | |
| **The nearest Health Service** | | | | | |
| Village maternity posts | 337 | 223 (66.2) | 2.02 (1.50, 2.72) | 1.93 (1.37, 2.70) | 0.000 |
| Village health Post | 226 | 165 (73.0) | 2.80 (1.97, 3.98) | 2.66 (1.80, 3.94) | 0.000 |
| Subsidiary Public Health centres | 313 | 187 (59.7) | 1.54 (1.14, 2.07) | 1.10 (0.79, 1.54) | 0.554 |
| Public Health centres | 411 | 202 (49.2) | 1.00 | 1.00 | |
| **Distance to the Nearest Health Facilities** | | | | | |
| < 1 Km | 484 | 295 (61.0) | 1.00 | 1.00 | |
| 1 - 2 Km | 360 | 204 (56.7) | 0.84 (0.64, 1.11) | 0.97 (0.71, 1.33) | 0.857 |
| 2 - 3 Km | 174 | 82 (47.1) | 0.57 (0.40, 0.81) | 0.63 (0.43, 0.93) | 0.019 |
| >=3 Km | 269 | 196 (72.9) | 1.72 (1.24, 2.38) | 1.73 (1.20, 2.50) | 0.003 |
| **†HH Income in relation to PMW** | | | | | |
| < PMW | 1151 | 733 (63.7) | 3.67 (2.51, 5.36) | 2.08 (1.33, 3.26) | 0.0014 |
| >= PMW | 136 | 44 (32.4) | 1.00 | 1.00 | |
| **‡HH Income in relation to IPL** | | | | | |
| < IPL | 1199 | 749 (62.5) | 3.57 (2.24, 5.67) | 0.79 (0.35, 1.81) | 0.585 |
| >= IPL | 88 | 28 (31.8) | 1.00 | 1.00§ | |

†The Provincial Minimum Wages (PMW) in 2019 is IDR 1,795,000 monthly; ‡The Indonesia Poverty Line (IPL) in 2019 is defined all households having the average income less than IDR 1,990,170 monthly; *Crude Odd Ratio (COR); **Adjusted Odd Ratio (AOR); §Variables are not significant at confidence interval (CI) 95%.

**Fig 4. The level of poor understanding of appropriate treatment-seeking behaviour of malaria by different sociodemographic and environmental characteristics in the East Nusa Tenggara Province, Indonesia (n = 777).**

The effectiveness of ACTs as the first-line treatment for malaria will occur when the treatment has been initiated within 24 hours of malaria [2]. This finding indicates that it is critical to improve the awareness of the rural population in this province to seek treatment promptly since delay in seeking treatment leads to adverse effects for both participants and the community. For an individual, seeking treatment after 24 hours increases their possibility for severe malaria anemia, blood transfusion [12], hospital inpatients [13], and mortality rate [59]. At the

community level, delay in seeking treatment increased the chance of onward transmission of the diseases to the community [11, 60].

Furthermore, the high proportion of rural adults with the perception of delay in seeking malaria treatment leads to ineffective treatment and management of subclinical malaria. Current literature indicates that prevalence of subclinical malaria at village level in the low endemic settings in this province varies ranging from 1.67% [61] to 13.23% [62] and the prevalence of asymptomatic malaria infection plasmodium vivax in asymptomatic malaria patients was higher compared with other plasmodiums in ENTP [63]. The treatment for plasmodium vivax needs fourteen days to obtain the optimal effect of the medicine [42]. Improving awareness at a community level for early detect of subclinical malaria is critical to progress to malaria elimination since subclinical malaria has great contribution to increase the burden of malaria in the community [64] including a chronic inflammation [65], maternal anemia and premature birth [66].

The study also shows that a high proportion of participant has a perception in seeking malaria treatment at non-health facilities. This finding was consistent with a study in India [67] and Bangladesh [68], however, it was contrasted with finding in Saudi Arabia [69] indicating that a high proportion of community to seek treatment at health facilities. The reason for this discrepancy might be due to rural community in Asia-Pacific lacking accessibility to health facilities as a consequence of financial issue [17]. Whilst, rural community in Saudi Arabia has increased accessibility to local health facilities and services [70]. Raising awareness at the community level to seek treatment at health facilities is crucial to ensure all clinical cases of malaria could be examined accurately under microscopies for allowing health practitioners to identify the types of malaria to provide treatment accordingly [71]. The study shows that participants with no education level were almost four times higher to seek malaria treatment at non-health facilities compared to those having a diploma or above. This is more likely that non educated people has low health literacy [72], therefore they might not know the right place to diagnosis their disease. This situation leads to the low rate in the use of professional health service as a highlight in other studies [73].

The present study demonstrated that the low level of education was significantly associated with the poor understanding of AMTSB. This study shows that adults with no education were three times higher to have poor understanding compared to adults with a diploma or above education level. These results were consistent with other studies in other countries such as Myanmar [46], India [48], and Northern Ethiopia [49]. This is more likely due to adults with a low level of education having poor knowledge on when and where to seek treatment for malaria since they have no ability to understand various written health information to improve their health behaviour. This situation worsened with decreased education levels with 34.2% of participants never hearing malaria term from the lowest level of education obtainment. Therefore, it could be assumed that most rural adults in this province have a poor understanding of AMTSB regardless of whether they hear or never heard of malaria term. This situation leads to worst the management of malaria treatment which causes by *Plasmodium* vivax which in turn hinders the effort to achieve malaria elimination in this province.

This study shows that distance to the nearest health facilities is one of the determinant factors associated with a poor understanding of AMTSB. The study has shown that the poor understanding of adults residing at least three kilometres from the nearest health services was almost two times higher compared to those living less than one kilometres. Our finding corroborates with the studies in Myanmar [15], Laos [74], India [13], Equatorial Guinea [51], and Tanzania [55]. This might be because people living far from health facilities were less exposed to malaria information since they had less access to health promotion activities. Literature

indicated health promotion workers were limited in this province [40], therefore malaria education campaign does not reach for those living far from health facilities in rural area [75].

This study shows that low social-economic status was significantly associated with poor understanding of AMTSB. The results indicated that rural adults living with low SES were almost two times more likely to have poor understanding compared with those from high SES. This finding was in line with study in Benin [50], Equatorial Guinea [51], and Nigeria [52]. A possible explanation could be people with low SES have poorer access to health facilities [76] and less exposure to multiple source of health information [77]. Furthermore, literature indicated public transports in village of ENTP were limited and people relied on walking by foot to accessing health facilities [78].

This study further shows that occupation was significantly associated with poor understanding of AMTSB. Rural adults working as farmers had poor understanding of AMTSB two times higher than those working in the public sector. This finding was consistent with studies in other settings [53, 54]. The reason for this could be that health literacy of farmer was lower than those working in public sectors [79] and that malaria awareness of farmer was lower than government and non-government workers in this province [80]. Education intervention throughout various types of media tailored to local aspect of community plays a significant role to improve health outcomes of farmers and their families as shown in other countries [81].

The study has identified the poor understanding of malaria treatment-seeking behaviour of rural communities in ENTP. Improving the awareness of this community to seek treatment effectively is fundamental to be addressed by the local authority to ensure universal access to malaria diagnosis and effective treatment, which is one of the global strategies recommended by WHO to achieve malaria elimination by 2030 [82]. Furthermore, this study has provided strong evidence that rural adults with a low level of education are the most vulnerable groups having a poor understanding of AMTSB. Considering of poor malaria awareness and the low level of education of the rural population in the ENTP [31, 36, 83, 84] special attention of the local authority should be focused on this vulnerable group to improve their awareness of AMTSB.

The strength of the study include that it was conducted based on a large sample size representing the rural population of three main ethnicities from three main islands of ENTP and three different malaria-endemic areas allowing authors to capture a different perspective of the local community on how their understanding of AMTSB. Furthermore, data collection supported by local nurses was conducted by face-to-face interview to accommodate the low literacy of the rural population in the province. Despite these, the study has several limitations. The treatment-seeking behaviour was assessed based on the hypothetical scenario. The participants were asked their opinion on when and where they would seek treatment if they or their family members experienced malaria. The hypothetical behaviour might be different from actual behaviour. However, this kind of treatment-seeking understanding is usually used to present variation of malaria treatment-seeking behaviour amongst demographic groups in the rural population. Furthermore, this study does not capture the effect of the availability of resources including instruments and microscopists to diagnose the malaria of local PHCs that might support treatment seeking behaviour of communities in the village and is an important area for further research.

## Conclusion

More than half of rural adults in this study have a poor understanding of appropriate malaria treatment-seeking behaviour. This situation leads to poor implementation of ACTs to treat

malaria and the completeness of malaria treatment. Rural adults having low-level education, low SES, farming occupation, and living at least three kilometres from the nearest health services were the most vulnerable groups that should be prioritized by policy health markers. The improvement of understanding of AMTSB for these groups will support the efficacy of malaria treatment in ENTP, as well as work to eliminate malaria by 2030 in this region. The capacity of human resources at local PHCs needs to be explored more to support AMTSB of rural communities in this province.

## Supporting information

**S1 Checklist. STROBE statement—checklist of items that should be included in reports of *cross-sectional studies*.**
(DOC)

**S1 Dataset. Database for study malaria treatment-seeking behaviour in rural East Nusa Tenggara Province, Indonesia.**
(PDF)

**S1 Questionnaire. The questionnaire for primary data collection.**
(PDF)

## Acknowledgments

We would like to thank to all participants contributing voluntary in this study. We further would like to express our gratitude to the Health Ministry of Indonesia, the governor of ENTP, head of East Sumba, Belu, and East Manggarai district, nine head of sub-districts, and forty-nine village leaders for allowing conducted this research in their region.

## Author Contributions

**Conceptualization:** Robertus Dole Guntur.

**Data curation:** Robertus Dole Guntur.

**Formal analysis:** Robertus Dole Guntur.

**Funding acquisition:** Robertus Dole Guntur.

**Investigation:** Robertus Dole Guntur.

**Methodology:** Robertus Dole Guntur.

**Project administration:** Robertus Dole Guntur.

**Resources:** Robertus Dole Guntur.

**Software:** Robertus Dole Guntur.

**Supervision:** Jonathan Kingsley, Fakir M. Amirul Islam.

**Validation:** Robertus Dole Guntur.

**Visualization:** Robertus Dole Guntur.

**Writing – original draft:** Robertus Dole Guntur.

**Writing – review & editing:** Robertus Dole Guntur, Jonathan Kingsley, Fakir M. Amirul Islam.

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
