## [Decision Letter · Decision Letter 0]

20 Oct 2021

PONE-D-21-17655Malaria treatment-seeking behaviour and its associated factors: A cross-sectional study in rural East Nusa Tenggara Province, IndonesiaPLOS ONE

Dear Dr. Guntur,

Thank you for submitting your manuscript to PLOS ONE. After careful consideration, we feel that it has merit but does not fully meet PLOS ONE’s publication criteria as it currently stands. Therefore, we invite you to submit a revised version of the manuscript that addresses the points raised during the review process.

We look forward to receiving your revised manuscript.

Kind regards,

Pyae Linn Aung, Ph.D.

Academic Editor

PLOS ONE

Journal Requirements:

2. Please include additional information regarding the survey or questionnaire used in the study and ensure that you have provided sufficient details that others could replicate the analyses. For instance, if you developed the survey or questionnaire as part of this study and it is not under a copyright more restrictive than CC-BY, please include a copy, in both the original language and English, as Supporting Information. If the questionnaire is published, please provide a citation to the (1) questionnaire and/or (2) original publication associated with the questionnaire."

3. Please include your actual numerical p-values in Figures 2-4 (tables)."

4. Please provide additional details regarding participant consent. In the ethics statement in the Methods and online submission information, please ensure that you have specified whether consent was informed.

“We would like to thank the Australia Awards Scholarship for supporting this research and the Faculty of Health, Arts, and Design of the Swinburne University Technology for providing funding for primary data collection for this study. The funders had no role in the designing of the study, data collection, analysis, or interpretation of data, or writing the paper. We further would like to express our gratitude to the Health Ministry of Indonesia, the governor of ENTP, head of East Sumba, Belu, and East Manggarai district, nine head of sub-districts, and forty-nine village leaders for allowing conducted this research in their region.”

Reviewers' comments:

Reviewer's Responses to Questions

**Comments to the Author**

1. Is the manuscript technically sound, and do the data support the conclusions?

Reviewer #1: Yes

Reviewer #2: Yes

2. Has the statistical analysis been performed appropriately and rigorously? 

Reviewer #1: Yes

Reviewer #2: Yes

3. Have the authors made all data underlying the findings in their manuscript fully available?

Reviewer #1: Yes

Reviewer #2: Yes

4. Is the manuscript presented in an intelligible fashion and written in standard English?

Reviewer #1: Yes

Reviewer #2: Yes

5. Review Comments to the Author

Reviewer #1: Malaria treatment-seeking behaviour and its associated factors: A cross-sectional study in rural East Nusa Tenggara Province, Indonesia

Abstract

Result

- Use the word "percent" after any number that begins a sentence, title or text heading. The numbers is that you should begin a sentence with a word even if the number is greater than nine, and the word "percent" should also be used. For example: Eighty-six percent of participants… (Edit throughout the whole manuscript)

- Low level of education or lower level of education?

- Kilometer should come up with full form in first time use instead of km

- Suggested to rephrase the last sentence of result in abstract : The style of interpretation is difficult to understand the results for the readers.

Introduction

- Types of delay can be classified into (1) Patient delay (2) Diagnosis delay (2) Treatment delay and (4) total delay. I think this study on malaria treatment-seeking behaviour of patients (Patient delay) and its associated factors. So, it should be explained more about determinants of patient delay in treatment seeking behaviours. In the introduction part, explanation about the reasons of delays is confuse.

Methods

- It is acceptable to refer the previous published manuscript regarding with the sample calculation and recruitment. But inclusion/exclusion of study participants should be illustrated again in this manuscript to easy understanding for the readers.

Results

- Begin a sentence with a word even if the number is greater than nine, and the word "percent" should also be used

- Better to include a statistical test result of P-value in the table

- Included study areas in the analysis represent for all the rural areas of Indonesia? Otherwise, it will limit for the study results in giving recommendation for the whole country.

Discussion

- The main factors associated with the poor understanding of the AMTSB were the low level of education, low social-economic status, working as housewife, and distance to the nearest health facilities. But authors fail to discuss covering all those main factors.

- Should discuss subclinical malaria and association with delay in treatment seeking behaviours

Conclusion

- better to be improved by adding some relevant recommendations or future studies, etc.

Reference

- Some references have inadequate formatting. Please go through all the references and edit as necessary.

Minor things

- There were a lot of textual and grammatical errors through the manuscript. Long sentences should be separated into short sentences to give better understanding for the readers.

Reviewer #2: Firstly, I would like to appreciate the authors for their satisfactory efforts. Almost all sections are well written and I have a few comments as follows:

1. Page 7 Sample size; the total sample size was 1495. The complete sample size calculation was referenced to the authors' previous paper. From my understandings, it was found that the number of samples was inconsistent. Please would you like to explain?

2. Page 17: second and third paragraphs; the authors mentioned that their findings were contrasted with findings in some countries. Would you like to add more explanations or suggestions for that point.

6. PLOS authors have the option to publish the peer review history of their article (what does this mean?). If published, this will include your full peer review and any attached files.

Reviewer #1: No

Reviewer #2: No

---

## [Author Response · Author response to Decision Letter 0]

29 Nov 2021

REBUTTAL LETTER

PONE-D-21-17655

Malaria treatment-seeking behaviour and its associated factors: A cross-sectional study in rural East Nusa Tenggara Province, Indonesia

PLOS ONE

Journal Requirements:

Response:

Thank you for this feedback. We have updated the manuscript to meet the PLOS ONE style requirements including file naming.

2. Please include additional information regarding the survey or questionnaire used in the study and ensure that you have provided sufficient details that others could replicate the analyses. For instance, if you developed the survey or questionnaire as part of this study and it is not under a copyright more restrictive than CC-BY, please include a copy, in both the original language and English, as Supporting Information. If the questionnaire is published, please provide a citation to the (1) questionnaire and/or (2) original publication associated with the questionnaire."

Response:

The English version of questionnaire has been published as a part of the protocol paper for this study and we have cited this publication as reference number 43. Whilst the original language version of the questionnaire has been attached as supporting information in this paper (page 23 line 22). 

3. Please include your actual numerical p-values in Figures 2-4 (tables).

Response:

Thank you so much for this feedback. The actual numerical p-value in Figure 2-4 has been included in all figures (Tables).

4. Please provide additional details regarding participant consent. In the ethics statement in the Methods and online submission information, please ensure that you have specified whether consent was informed.

Response:

Thank you for this feedback. The participant consent has been specified in the ethical consideration section as: 

“All participants provided written consent prior interview being conducted” (page 12, line 6 and 7). 

“We would like to thank the Australia Awards Scholarship for supporting this research and the Faculty of Health, Arts, and Design of the Swinburne University Technology for providing funding for primary data collection for this study. The funders had no role in the designing of the study, data collection, analysis, or interpretation of data, or writing the paper. We further would like to express our gratitude to the Health Ministry of Indonesia, the governor of ENTP, head of East Sumba, Belu, and East Manggarai district, nine head of sub-districts, and forty-nine village leaders for allowing conducted this research in their region.”

Response:

Thank you for this feedback. We have updated the acknowledgment section as below:

”We would like to thank to all participants contributing voluntary in this study. We further would like to express our gratitude to the Health Ministry of Indonesia, the governor of ENTP, head of East Sumba, Belu, and East Manggarai district, nine head of sub-districts, and forty-nine village leaders for allowing conducted this research in their region” (page 23, line 13 – 16).

Reviewers' comments:

Reviewer #1: Malaria treatment-seeking behaviour and its associated factors: A cross-sectional study in rural East Nusa Tenggara Province, Indonesia

Abstract

Result

1. Use the word "percent" after any number that begins a sentence, title or text heading. The numbers is that you should begin a sentence with a word even if the number is greater than nine, and the word "percent" should also be used. For example: Eighty-six percent of participants… (Edit throughout the whole manuscript)

Response:

Thank you for this feedback. We have improved abstract as below:

“Eighty-six percent of participants were found to be familiar with the term malaria” (page 2, line 17). We have also made adjustments to the whole manuscript regarding this issue. 

2. Low level of education or lower level of education?

Response:

We have improved the result section of the abstract and the statement “lower level of education” disappear from this section.

3. Kilometer should come up with full form in first time use instead of km

Response:

We have improved abstract as below: 

“residing at least three kilometres (km) away from the nearest health facilities (AOR: 1.73, 95% CI: 1.2, 2.5) compared to those living within one km from the nearest health service” (page 2, line 22 and 23).

4. Suggested to rephrase the last sentence of result in abstract: The style of interpretation is difficult to understand the results for the readers.

Response:

We have improved the last sentence of the result in abstract as below: 

“Whilst, other factors such as ethnicity and family size were not associated with the poor understanding of AMTSB” (page 2, line 25 and page 3 line 1). 

The style of interpretation of the results has been improved as below: 

“Poor understanding of AMTSB was significantly higher for adults with no education (adjusted odds ratio (AOR) 3.42, 95% CI: 1.81, 6.48) compared to those with a diploma or above education level; having low SES (AOR: 1.87, 95% CI: 1.19, 2.96) compared to those having high SES; residing at least three kilometres (km) away from the nearest health facilities (AOR: 1.73, 95% CI: 1.2, 2.5) compared to those living within one km from the nearest health service; and working as farmer (AOR: 1.63, 95% CI: 1.01 – 2.63) compared to those working at government or non-government sector” (page 2 line 19 -25). 

Introduction

1. Types of delay can be classified into (1) Patient delay (2) Diagnosis delay (2) Treatment delay and (4) total delay. I think this study on malaria treatment-seeking behaviour of patients (Patient delay) and its associated factors. So, it should be explained more about determinants of patient delay in treatment seeking behaviours. In the introduction part, explanation about the reasons of delays is confuse.

Response:

Thank you for this feedback. We have updated the introduction section as below:

“Moreover, studies on various settings among Asia communities indicated that patients sought care for malaria at health facilities on average at least two days after the onset of symptoms [13-16]. One major reason of delaying in seeking malaria treatment of patients in rural communities of low to middle-income Asia-Pacific countries was the use of traditional medicine (TM) [17]. The prevalence of the application of TM as the first option before visiting health facilities in rural settings of the region varied from 2.5% to 40%. Other reasons for this included difficulty in accessing health facilities [16], living far from health facilities [13,15,16], and practicing self-treatment by buying drugs at local shops [14]” (page 4, line 13 – 21).

“The high coverage of ACTs in Indonesia, however, is hampered by the behavior of people who treat their malaria without consultation from health practitioners and delay in seeking care at health facilities. The prevalence of self-medication for malaria treatment among the Indonesian population accounted for 0.6% of the populace, with variation from 0.2% in Bali Province to 5.1% in West Papua Province [21]. Furthermore, most of malaria patients in some parts of Indonesia visited health facilities after three days of the onset of symptoms [24-26] and the use of TM as the first option for malaria treatment leads to delay for several days in visiting health services [27,28]. However, the associated factors related to delay in malaria treatment seeking behaviour of the patients were not investigated in aforementioned studies” (page 5, line 8 -16). 

Methods

1. It is acceptable to refer the previous published manuscript regarding with the sample calculation and recruitment. But inclusion/exclusion of study participants should be illustrated again in this manuscript to easy understanding for the readers.

Response:

Thank you for this feedback. We have updated the recruitment section as below:

“One household head was selected as study participant in the visited households. Any household head whether male or female who wanted to participate had to provide a voluntary consent form and be over the age of 18 years to be included in this study. In cases when there were two or more household heads in single dwelling, the household head occupying firstly on that home was selected as participants. Since we want to explore malaria treatment seeking behavior of rural adults in ENTP, we exclude any household heads who were less than 18 years of age. This criterion was based on the guidance of the ethic of this research” (page 8, line 15 -22). 

Results

1. Begin a sentence with a word even if the number is greater than nine, and the word "percent" should also be used

Response:

All sentences started with number have been updated by saying the number in word as below: 

“Thirteen point nine percent (208 out of 1495) of participants have never heard of malaria” (page 12, line 13-14). 

“Seventy one percent of participants with poor SES compared to 42.8% of participants with rich SES were found to have a poor understanding of AMTSB” (page 16, line 24 and 25 and page 17 line 1).

2. Better to include a statistical test result of P-value in the table

Response:

All tables have been supported with the statistical test result of P-value as indicated in Figure 2, Figure 3 and Figure 4. 

3. Included study areas in the analysis represent for all the rural areas of Indonesia? Otherwise, it will limit for the study results in giving recommendation for the whole country.

Response:

Thank you for this feedback. The rural area in this study represents the rural area in ENTP of Indonesia. 

“Perception of time and place to seek malaria treatment of rural adults of ENTP is presented in Table 2” (page 13, line 2 and 3).

“After adjustment for gender and age group, the following variables were significantly associated with the poor understanding of AMTSB for rural adults in ENTP” (page 17, line 5 and 6). 

Therefore, we provided our recommendation for this province as below: 

“The study has identified the poor understanding of malaria treatment-seeking behaviour of rural communities in ENTP. Improving the awareness of this community to seek treatment effectively is fundamental to be addressed by the local authority to ensure universal access to malaria diagnosis and effective treatment, which is one of the global strategies recommended by WHO to achieve malaria elimination by 2030 [82]” (page 21, line 22 – 25 and page 22 line 1). 

“The improvement of understanding of AMTSB for these groups will support the efficacy of malaria treatment in ENTP, as well as work to eliminate malaria by 2030 in this region” (page 23, line 7 and 8). 

Discussion

1. The main factors associated with the poor understanding of the AMTSB were the low level of education, low social-economic status, working as housewife, and distance to the nearest health facilities. But authors fail to discuss covering all those main factors.

Response:

Thank you for this feedback. We have improved the article as below:

“The main factors associated with the poor understanding of the AMTSB were the low level of education, low social-economic status, farming occupation, and distance to the nearest health facilities” (page 18, line 2 - 4).

The discussion of low level of education and distance to the nearest health facilities in relation with the poor understanding of AMTSB has been described previously. We have updated the article with two paragraphs for the discussion of low social-economic status and farming occupation in relation with the poor understanding of AMTSB as below:

“This study shows that low social-economic status was significantly associated with poor understanding of AMTSB. The results indicated that rural adults living with low SES were almost two times more likely to have poor understanding compared with those from high SES. This finding was in line with study in Benin [50], Equatorial Guinea[51], and Nigeria [52]. A possible explanation could be people with low SES have poorer access to health facilities [76] and less exposure to multiple source of health information [77]. Furthermore, literature indicated public transports in village of ENTP were limited and people relied on walking by foot to accessing health facilities [78]” (page 21, line 4 – 11). 

“This study further shows that occupation was significantly associated with poor understanding of AMTSB. Rural adults working as farmers had poor understanding of AMTSB two times higher than those working in the public sector. This finding was consistent with studies in other settings [53,54]. The reason for this could be that health literacy of farmer was lower than those working in public sectors [79] and that malaria awareness of farmer was lower than government and non-government workers in this province [80]. Education intervention throughout various types of media tailored to local aspect of community plays a significant role to improve health outcomes of farmers and their families as shown in other countries [81]” (page 21, line 13 – 20). 

2. Should discuss subclinical malaria and association with delay in treatment seeking behaviours.

Response:

Thank you for this feedback. We have added a paragraph talking about subclinical malaria and the association with delay in treatment seeking behaviour as below: 

“Furthermore, the high proportion of rural adults with the perception of delay in seeking malaria treatment leads to ineffective treatment and management of subclinical malaria. Current literature indicates that prevalence of subclinical malaria at village level in the low endemic settings in this province varies ranging from 1.67% [61] to 13.23% [62] and the prevalence of asymptomatic malaria infection plasmodium vivax in asymptomatic malaria patients was higher compared with other plasmodiums in ENTP [63]. The treatment for plasmodium vivax needs fourteen days to obtain the optimal effect of the medicine [42]. Improving awareness at a community level for early detect of subclinical malaria is critical to progress to malaria elimination since subclinical malaria has great contribution to increase the burden of malaria in the community [64] including a chronic inflammation [65], maternal anemia and premature birth [66]” (page 19, line 1 – 11). 

Conclusion

1. Better to be improved by adding some relevant recommendations or future studies, etc.

Response:

Thank you for this feedback. The last part of the conclusion section has been updated as below: 

“The improvement of understanding of AMTSB for these groups will support the efficacy of malaria treatment in ENTP, as well as work to eliminate malaria by 2030 in this region. The capacity of human resources at local PHCs needs to be explored more to support AMTSB of rural communities in this province” (page 23, line 7 – 10). 

Reference

1. Some references have inadequate formatting. Please go through all the references and edit as necessary.

Response:

All the references have been updated following the guidance of the PLOS ONE journal.

Minor things

1. There were a lot of textual and grammatical errors through the manuscript. Long sentences should be separated into short sentences to give better understanding for the readers.

Response:

Thank you for this feedback. The manuscript has been updated and proofread by the second author who is an English native speaker.

Reviewer #2: Firstly, I would like to appreciate the authors for their satisfactory efforts. Almost all sections are well written and I have a few comments as follows:

1. Page 7 Sample size; the total sample size was 1495. The complete sample size calculation was referenced to the authors' previous paper. From my understandings, it was found that the number of samples was inconsistent. Please would you like to explain?

Response:

Thank you for this feedback. We have updated the sample size as below:

“The total sample recruited for the study was 1503 adults. This number was obtained after considering the prevalence of malaria in ENTP, the intra-class coefficient correlation for malaria prevention study in Indonesia, design effect, and participation rate of participants. The complete sample size calculation was presented previously [43]” (page 8, line 2-5).

2. Page 17: second and third paragraphs; the authors mentioned that their findings were contrasted with findings in some countries. Would you like to add more explanations or suggestions for that point?

Response:

Thank you for this feedback. We have updated the second and third paragraphs of the discussion section as below:

“This study shows that more than half of participants delay in seeking treatment for their malaria. This finding is consistent with studies in Myanmar [46] and Nigeria [47]. However, our finding contrasted with studies in Cabo Verde [57] and South Africa [58], revealing that the high awareness of the community to seek treatment within 24 hours. This discrepancy might be due to different level of malaria knowledge of communities in those settings. Rural communities in ENTP Indonesia [36], Myanmar [46] and Nigeria [47] has low malaria knowledge, whilst rural adults in Cabo Verde and South Africa have high malaria knowledge leads to timely seek malaria treatment [57,58]. The low awareness to seek treatment within 24 hours leads to ineffective implementation of ACTs as the first line of malaria treatment, even though the coverage of ACTs in ENTP has increased from 55% in 2013 [21] to 93.9% in 2020 [22]” (page 18, line 6 -15).

“The study also shows that a high proportion of participant has a perception in seeking malaria treatment at non-health facilities. This finding was consistent with a study in India [67] and Bangladesh [68], however, it was contrasted with finding in Saudi Arabia [69] indicating that a high proportion of community to seek treatment at health facilities. The reason for this discrepancy might be due to rural community in Asia-Pacific lacking accessibility to health facilities as a consequence of financial issue [17]. Whilst, rural community in Saudi Arabia has increased accessibility to local health facilities and services [70]. Raising awareness at the community level to seek treatment at health facilities is crucial to ensure all clinical cases of malaria could be examined accurately under microscopies for allowing health practitioners to identify the types of malaria to provide treatment accordingly [71]” (page 19, line 13 – 22).

---

## [Decision Letter · Decision Letter 1]

14 Jan 2022

Malaria treatment-seeking behaviour and its associated factors: A cross-sectional study in rural East Nusa Tenggara Province, Indonesia

PONE-D-21-17655R1

Dear Dr. Guntur,

We’re pleased to inform you that your manuscript has been judged scientifically suitable for publication and will be formally accepted for publication once it meets all outstanding technical requirements.

Kind regards,

Pyae Linn Aung, Ph.D.

Academic Editor

PLOS ONE

Additional Editor Comments (optional):

I have no special comments regarding the revised manuscript. Thank authors for addressing all the reviewers comments.

Reviewers' comments:

Reviewer's Responses to Questions

**Comments to the Author**

1. If the authors have adequately addressed your comments raised in a previous round of review and you feel that this manuscript is now acceptable for publication, you may indicate that here to bypass the “Comments to the Author” section, enter your conflict of interest statement in the “Confidential to Editor” section, and submit your "Accept" recommendation.

Reviewer #1: All comments have been addressed

Reviewer #2: (No Response)

2. Is the manuscript technically sound, and do the data support the conclusions?

Reviewer #1: Partly

Reviewer #2: Yes

3. Has the statistical analysis been performed appropriately and rigorously? 

Reviewer #1: Yes

Reviewer #2: Yes

4. Have the authors made all data underlying the findings in their manuscript fully available?

Reviewer #1: Yes

Reviewer #2: Yes

5. Is the manuscript presented in an intelligible fashion and written in standard English?

Reviewer #1: Yes

Reviewer #2: Yes

6. Review Comments to the Author

Reviewer #1: The reviewer would like to thank the authors for addressing the comments.

The authors have largely addressed the detailed questions I raised in my previous review.

The revised manuscript is well-written, with precise terminology and detailed descriptions of the methodology,

data analysis and discussion.

Reviewer #2: (No Response)

7. PLOS authors have the option to publish the peer review history of their article (what does this mean?). If published, this will include your full peer review and any attached files.

Reviewer #1: No

Reviewer #2: No

---

## [Editor Report · Acceptance letter]

26 Jan 2022

PONE-D-21-17655R1 

Malaria treatment-seeking behaviour and its associated factors: A cross-sectional study in rural East Nusa Tenggara Province, Indonesia 

Dear Dr. Guntur:

I'm pleased to inform you that your manuscript has been deemed suitable for publication in PLOS ONE. Congratulations! Your manuscript is now with our production department. 

Kind regards, 

on behalf of

Dr. Pyae Linn Aung 

Academic Editor

PLOS ONE